# Prenatal Exome Sequencing in Recurrent Fetal Structural Anomalies: Systematic Review and Meta-Analysis

**DOI:** 10.3390/jcm10204739

**Published:** 2021-10-15

**Authors:** Montse Pauta, Raigam Jafet Martinez-Portilla, Antoni Borrell

**Affiliations:** 1BCNatal, Institut d’Investigacions Biomèdiques August Pi i Sunyer (IDIBAPS), 08036 Barcelona, Catalonia, Spain; mpauta@clinic.cat (M.P.); raifet@hotmail.com (R.J.M.-P.); 2Clinical Research Division, Evidence-Based Medicine Department, National Institute of Perinatology, Mexico City 01120, Mexico; 3BCNatal, Department of Obstetrics and Gynecology, Hospital Clínic de Barcelona, 08028 Barcelona, Catalonia, Spain; 4Medical School, University of Barcelona, 08036 Barcelona, Catalonia, Spain

**Keywords:** exome sequencing, diagnostic yield, prenatal diagnosis, fetal structural anomalies, recurrent anomalies

## Abstract

To determine the diagnostic yield of exome sequencing (ES), a microarray analysis was carried out of fetuses with recurrent fetal structural anomalies (with similar anomalies in consecutive pregnancies). This is a systematic review conducted in accordance with Preferred Reporting Items for Systematic Reviews and Meta-Analyses (PRISMA) criteria. The selected studies describing ES in fetuses with recurrent fetal malformation were assessed using the Standards for Reporting of Diagnostic Accuracy Studies (STARD) criteria for risk of bias. Incidence was used as the pooled effect size by single-proportion analysis using random-effects modeling (weighted by inverse of variance). We identified nine studies on ES diagnostic yield that included 140 fetuses with recurrent structural anomalies. A pathogenic or likely pathogenic variant was found in 57 fetuses, resulting in a 40% (95%CI: 26% to 54%) incremental performance pool of ES. As expected, the vast majority (86%: 36/42) of the newly identified diseases had a recessive inheritance pattern, and among these, 42% (15/36) of variants were found in homozygosity. Meckel syndrome was the monogenic disease most frequently found, although the genes involved were diverse. The ES diagnostic yield in pregnancies with recurrent fetal structural anomalies was 40% (57/140). Homozygous disease-causing variants were found in 36% (15/57) of the newly identified monogenic disorders.

## 1. Introduction

Structural fetal anomalies occur in approximately 2.5% of pregnancies [1], and at present most of them can be identified by ultrasound examination during pregnancy. Classically, when a fetal structural anomaly was detected prenatally, fetal karyotyping was able to reveal a chromosomal anomaly in 14% of the cases [2]. Currently, a chromosomal microarray analysis is preferred because it provides up to a 6–10% incremental diagnostic yield above the fetal karyotype [3,4]. Sometimes, in a subsequent pregnancy a similar congenital anomaly can be observed. The frequency of recurrent fetal anomalies in singleton pregnancies of the general population has been reported to be 0.03% (301/872, 493) by the Northern England Congenital Abnormality Survey [5].

Exome sequencing (ES) enables the assessment of the coding regions of more than 20,000 genes, which comprise approximately 1 to 2% of the genome. Nowadays, ES is increasingly being used prenatally in fetuses with structural anomalies and normal microarrays. To simplify interpretation and minimize inconclusive findings, the analysis can be restricted to the coding sequences of the OMIM (Online Mendelian Inheritance in Man) genes (clinical or medical ES), or to those genes previously described to be associated with a specific condition (gene panels), instead of interpreting the whole exome (whole ES). Studies using whole-genome sequencing (GS) are still scarce in the prenatal diagnosis field.

It has been shown that, for women whose first pregnancy was affected by a fetal structural anomaly, the absolute recurrence risk of a similar anomaly in the second pregnancy is 2%, resulting in a 4.08% overall risk of congenital anomalies when an extra 2% risk for dissimilar anomalies is added. Regarding the recurrence risk (RR) for a similar anomaly, it was considerably elevated (RR = 24) [5]. This recurrence risk increases dramatically when an autosomal recessive or an X-linked disease is demonstrated by molecular analysis.

In this study, we aim to perform a systematic review of the literature and meta-analysis to assess the diagnostic yield of ES in fetuses with recurrent structural anomalies and a negative result at microarray or karyotyping.

## 2. Methods

### 2.1. Protocol and Registration

The protocol of this systematic review and meta-analysis on prenatal ES and GS in recurrent fetal structural anomalies was prospectively registered and published in an international open-access database for prospective protocols and can be accessed in the following link (http://doi.org/10.17605/OSF.IO/PDBCZ accessed on 23 April 2021). There is no need for institutional approval in our hospital for systematic reviews and meta-analysis. This meta-analysis adhered to the Preferred Reporting Items for Systematic Reviews and Meta-Analyses (PRISMA) guidelines for randomized controlled trials. The study protocol was agreed to among the authors before running the analysis, and one of them (R.M.), being external to the group, acted as a reviewer.

### 2.2. Eligibility Criteria

Criteria for inclusion in this systematic review were observational studies on pregnancies with the following inclusion criteria: (a) fetuses presenting a recurrent fetal structural anomaly (similar anomalies in consecutive pregnancies); (b) absence of a known familial mutation; (c) negative microarray result, treated as the reference standard; and (d) series with more than two cases in the English language. Positive variants classified as IV or V (likely pathogenic or pathogenic) were determined to be causative of the fetal phenotype. Therefore, variants of uncertain significance (VUS) and secondary findings were not extracted. Either ES or GS applied as a solo (the fetus alone) or trio (fetus and both parents) approach were included. The following studies were excluded: (a) case reports; (b) opinion articles or letters; (c) application of gene panels; (d) studies dealing with specific syndromes or malformations; and (e) when data could not be extracted, and the corresponding author did not provide additional information.

### 2.3. Information Sources and Search

A systematic search was conducted using PubMed, SCOPUS, the Web of Knowledge, and the Cochrane database to identify relevant manuscripts published without time limits. References of relevant publications were manually searched for any additional potentially relevant studies that had been published. The first search was run on 15 November 2020, although an update was extended until 20 March 2021. The following MeSH terms (The Medical Subject Headings) with word variation of “genome” and “exome sequencing” and “fetuses” were used in an attempt to capture as many relevant studies as possible. Terms for “exome sequencing” include “genome”, “exome”, “exome sequencing”, or “whole-exome”, and alternative terms for “fetuses” included “fetal”, “prenatal diagnosis”, and “recurrent fetal malformation”.

### 2.4. Study Selection

Abstracts identified as relevant were assessed by two independent evaluators (M.P. and A.B.). If studies complied with the inclusion criteria, full-text articles were reviewed. In cases of relevant studies with missing information, the corresponding authors were contacted by e-mail. The search strategy and query syntaxes are depicted in Appendix A.

### 2.5. Data Collection Process and Data Items

The following data were extracted onto a datasheet based on a Cochrane Consumers and Communication Review Group data extraction template: countries where the study was carried out, study period, study inclusion criteria, sample size, number of cases with recurrent fetal malformation, sequencing approach, Sanger validation, and results of ES/GS. In the case of a positive result, gene, type of variant, classification of variant, and syndrome or disease caused were extracted.

### 2.6. Outcome Measures

The primary outcome of the study was the diagnostic yield with ES/GS among fetuses with recurrent fetal structural anomalies, unknown Mendelian inheritance, and normal microarray. Diagnostic yield was defined as the number of positive ES/GS results among fetuses with recurrent fetal anomalies over the total number of cases with recurrent fetal anomalies (positive plus negative cases).

### 2.7. Assessment of Risk of Bias

Quality assessment was performed using modified Standards for Reporting of Diagnostic Accuracy (STARD) criteria [6]. The quality criteria deemed most important to optimize accuracy were (a) eligibility criteria described (consecutive or not) and the description or ES/GS approach; (b) specific phenotype study of the recurrent fetal structural anomaly; (c) in depth description of the fetal structural anomaly by ultrasound, postmortem studies, or neonatal examination; and (d) American College of Medical Genetics and Genomics (ACMG) classification used. Each parameter was graded as high, unclear, or low risk of bias. The risk of bias was measured individually by two reviewers (M.P. and A.B.).

### 2.8. Strategy for Data Synthesis and Statistical Analysis

The results extracted were pooled in a meta-analysis. For the primary outcome, incidence was used as the pooled effect size by single-proportion analysis [7] using random-effects modeling (weighted by inverse of variance), along with the Clopper–Pearson exact method for calculation of confidence intervals [8]. We used single proportion analysis to show the proportion of positive ES/GS among fetuses with recurrent fetal anomalies over the total number of fetuses with recurrent fetal anomalies. This analysis requires only cases among the total population to assess the proportion, which is interpreted as the diagnostic yield. Between-study heterogeneity/variability was assessed using the Τau^2^, Χ^2^ (Cochrane Q), and I^2^ statistics. Results were assessed using forest plots and presented as proportions. Publication bias was visually assessed by funnel plots [9,10], quantified by the Egger method (weighted linear regression of the treatment effect on its standard error) [11], and adjusted using the Copas model for selection bias [12,13,14]. Statistical analyses were conducted using R studio v1.0.136 (The R Foundation for Statistical Computing; (Boston MA, “meta” package for meta-analysis) package “meta v4.2”) [15].

## 3. Results

### 3.1. Study Selection and Study Characteristics

For the scoping review selection progress, 106 studies were initially selected from PubMed focusing on ES/GS and fetuses with a recurrent malformation. After full review, one study was deemed eligible and was included in this review [16]. Studies dealing with specific syndromes or malformations were excluded [17,18]. During the interactive review, we found series of structurally abnormal fetuses that included cases with a recurrent anomaly, and only those with more than two cases were considered (Figure 1). After full-text reviewing 29 articles, eight ES studies including fetuses with a recurrent structural anomaly were included [19,20,21,22,23,24,25]. Two articles with less than two cases of recurrent malformations [26,27], and one in which data extraction was not feasible [27], were excluded. No series on GS were found. Finally, data obtained from our own center and previously presented at international conferences were added [28].

A single study was focused on recurrent anomalies [16], while the remaining eight also included anomalous fetuses with non-recurrent anomalies (the latter were not included in the study). All studies were of high quality according to modified Standards for Reporting of Di-agnostic Accuracy (STARD) criteria (Figure 2). These studies were performed in centers from seven different countries (three from USA and one from each of the following countries: China, Greece, Switzerland, Netherlands, Israel, and Catalonia, Spain (Table 1)).

### 3.2. Risk of Bias of Included Studies

The quality of the studies was assessed using a modified Standards for Reporting of Diagnostic Accuracy (STARD) for this project. We considered the following quality criteria: (a) well defined inclusion criteria; (b) phenotype correctly described; (c) study focused on fetuses with recurrent structural anomalies; (d) genetic variants classified according the ACMG; (e) more than five fetuses included; (f) subsequent Sanger validation; (g) VUS and incidental findings reported; and (h) same previous genetic test applied to all fetuses.

### 3.3. Diagnostic Yield in Recurrent Malformation

Among the nine studies included, the risk of variants likely to be pathogenic or being pathogenic among fetuses with recurrent structural anomalies and normal results at microarray or karyotyping, by the random effects model, was 40% (95%CI: 26% to 54%) (Figure 3). The diagnostic yield observed in each of the included studies ranged from 12% to 67%, being 60% in the larger study with 40 cases. Heterogeneity was due to sample error rather than a true-effect, and according to the I^2^, the proportion of heterogeneity was 62%. Publication bias by the linear regression asymmetry test showed no significant quantification of bias (bias: 1.529; *p* = 0.059), as empirically depicted by the funnel plot (Appendix A).

When the 57 positive cases were grouped according the anatomic systems that were involved in the recurrent structural anomalies, the most commonly observed pattern was multisystem anomalies (*n* = 31), followed by central nervous system (*n* = 10), hydrops (*n* = 7), musculoskeletal (*n* = 7), and cardiac anomalies (*n* = 2).

### 3.4. Monogenic Variants

Overall, 87 causative variants of the phenotype were described in 57 of the 140 fetuses with recurrent anomalies. In 45 cases (79%), causative variants were found in genes with an autosomal recessive inheritance pattern; in six in recessive X-linked genes and inherited from a carrier mother (two cases of *L1CAM*, *FOXP3* and one case of *OFD1*, *AMER1*); two in dominant X-linked genes (*NONO*, *ZRSR2*); and finally, in four cases in dominant inheritance genes (three “de novo” and one maternal germinal mosaicism (*TP63*)). Among the 45 cases described with autosomal recessive inheritance, homozygosity was identified in 13 (29%) cases, although you must take into account that the largest study, including half of the identified variants, only enrolled non-consanguineous couples [16] (Table 2).

### 3.5. Associated Genes and Fetal Structural Anomalies

The syndrome most frequently found was Meckel syndrome, diagnosed in five fetuses although caused by four different genes (*CEP290* in two cases and *CC2D2A*, *TCTN2*, and *MSK1* in one case each) (Table 2). Four genes were found to be involved in four cases each: the *RAPSN* gene related to fetal akinesia syndrome II [16,19], the *SLC26A3* gene related to congenital chloride diarrhea [16], the *L1CAM* gene related to hydrocephalus with X-linked inherence [16,20], and the *FOXC2* gene related to lymphedema-distichiasis syndrome [25]. Among the 57 fetuses with a positive diagnosis, in 26 (46%) there was a single anatomical system involved, while in the remaining 31 there was more than one. The anatomical systems more frequently involved in structural anomalies were the following: central nervous system (*n* = 24), musculoskeletal (including polydactyly and arthrogryposis) (*n* = 20), hydrops (*n* = 14), nephrourological (*n* = 13), and cardiovascular (*n* = 10).

## 4. Discussion

Our systematic review and meta-analysis of ES/GS in fetuses with a recurrent structural anomaly, a normal result at microarray, and no family disease or Mendelian inheritance identified revealed a 40% (95%CI: 26%–54%) diagnostic yield. In most of the positive cases (79%), an autosomal recessive inheritance was identified, although homozygous variants were only found in 29% of those cases. The high incidence of autosomal recessive diseases was expected since this review includes recurrent fetal malformations from healthy parents. This incidence is expected to depend on the degree of consanguinity and the frequency of autosomal recessive carriers of a given population. Another factor that can explain the wider range of diagnostic yields (12% to 67%) observed among the included series is the different type of recurrent structural anomalies included. Multisystem anomalies appear to carry a higher diagnostic yield since they account for more than half of the positive cases.

Meckel syndrome was the most frequently found monogenic disease in this systematic review and accounted for five of the 57 cases reported. Meckel syndrome (OMIM PS249000), also known as Meckel–Gruber syndrome, is a lethal autosomal recessive syndrome that represents the most severe condition in a group of disorders collectively termed the ciliopathies. Meckel syndromre is characterized by the triad of cystic renal disease, posterior fossa abnormalities (usually occipital encephalocele), and hepatic ductal plate malformation, leading to hepatic fibrosis and bile duct proliferation. Polydactyly is relatively common. The prevalence is estimated at 1/38,500 births in Europe [29], although it can be higher in specific populations with a high consanguinity rate such as Gujarati Indians (1/1300) or Qatar (1/5000) [30]. A large European series with 191 Meckel syndrome cases noted that 90% of them were diagnosed prenatally at a mean gestational age of 14.3 (range 11–36) weeks [29]. Hence, the reason for the high representation of Meckel syndrome in this review may be its characteristic ultrasound pattern, which may facilitate an easy prenatal recognition rather than a high prenatal prevalence [31].

A large prospective study carried out in the Spanish population demonstrated the frequency of carriers of prevalent diseases. The autosomal recessive diseases reported with a carrier frequency higher than 1/40 were the following: *GJB2*-related DFNB1 non-syndromic hearing loss and deafness, cystic fibrosis, alpha-thalassemia, phenylketonuria, spinal muscular atrophy, familial Mediterranean fever, and autosomal recessive polycystic kidney disease (ARPKD) [31]. Although prevalent in this population, they have not been described in this review because affected fetuses do not typically present with structural anomalies. Although autosomal dominant polycystic kidney disease can be easily recognized by ultrasound, prenatal diagnosis of ARPKD is currently unreliable. Preconceptional or prenatal screening of carriers of autosomal recessive and X-linked diseases can identify potential affected fetuses irrespective of their ultrasound expressivity. The reason why carrier screening in Europe and other world regions is only offered to couples conceiving by assisted reproduction techniques is unclear.

Currently, ES is being increasingly applied to prenatal diagnosis, as “whole ES” when all the exons are studied, “clinical ES” when only the exons of the OMIM genes are interpreted, or even as “gene panel” when specific genes are selected according to the fetal phenotype. Gene panels were not included in this study, although skeletal, hydrops, or nephrourological panels could be helpful in fetuses with structural anomalies involving a single anatomical system, which in this review account for half of the positive cases. Among the nine studies included, five used clinical ES and four whole ES, and the trio approach was the most used. There is strong evidence of ES being a powerful tool in discovering the underlying etiology of recurrent fetal malformations detected by prenatal ultrasonography. Expansion from ES that only covers mutations in coding regions (~97% of exons) to GS, which also covers noncoding regions of the genome in the near future, is expected to increase the diagnostic yield. It cannot reliably detect copy-number variants at the single gene level.

A fact that makes the prenatal identification of monogenic syndromes difficult is the discordance between the fetal phenotype and the pediatric or adult phenotype, and this is not well described, yet. The authors of the largest study included in this review, Guo et al., reported four genes (*PUS3, SZT2, LAMA5,* and *ZRSR2*) that were found to show discordant phenotypes between the prenatal and postnatal periods, suggesting new relationships between genes and phenotypes in the fetal stage, which may expand the spectrum of the disease [16].

## 5. Conclusions

This systematic review and meta-analysis demonstrate that the ES technique is particularly useful when applied to prenatal diagnosis of monogenic syndromes, due to the lack of well-understood phenotypes in this early stage of human development. ES has been shown to improve the diagnostic yield of recurrent fetal structural anomalies, to be able to discover new genes potentially relevant in human development, and to be able to identify new pathogenic variants.

## Figures and Tables

**Figure 1 jcm-10-04739-f001:**
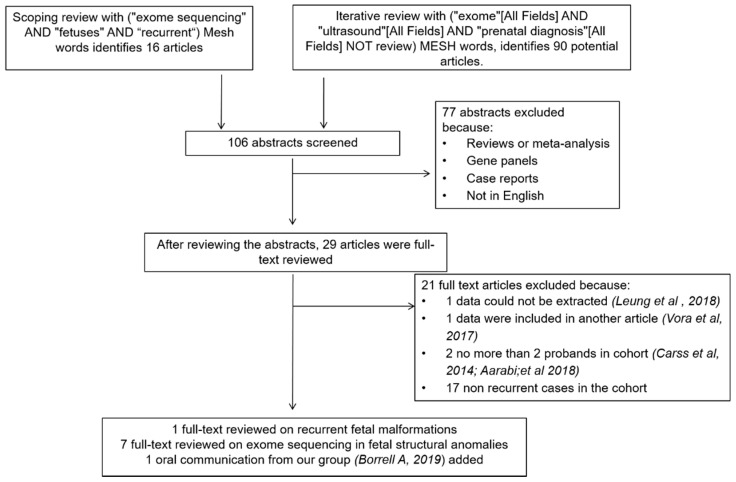
Flowchart summarizing inclusion in this systematic review of studies reporting on diagnostic yield of exome sequencing in fetuses with recurrent structural anomalies and a negative chromosomal microarray analysis or karyotyping, and no family history.

**Figure 2 jcm-10-04739-f002:**
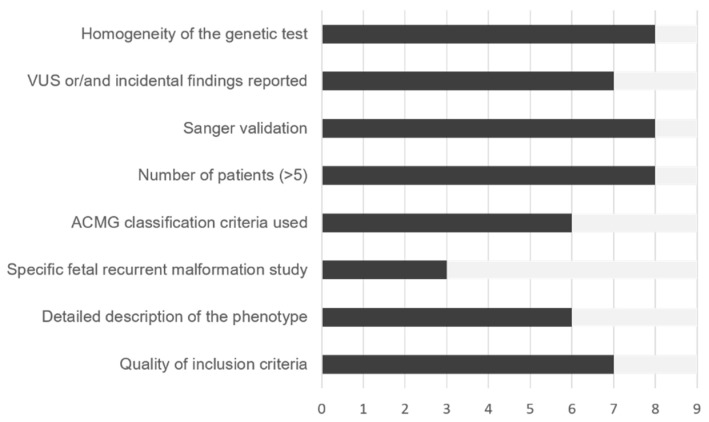
Quality assessment of the nine studies included in this systematic review using eight criteria. VUS: A variant of uncertain (or unknown) significance; ACMG: American College of Medical Genetics and Genomics.

**Figure 3 jcm-10-04739-f003:**
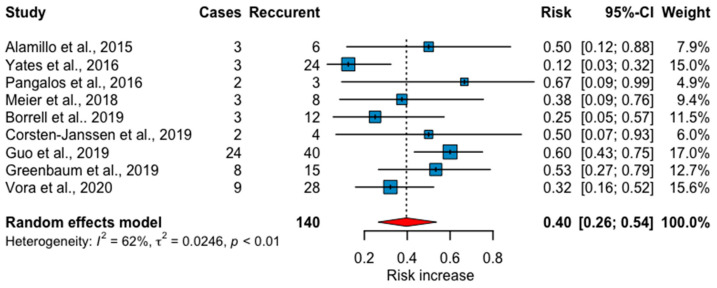
Forest plot of the diagnostic yield of exome sequencing in 140 fetuses with recurrent structural anomalies from nine studies.

**Table 1 jcm-10-04739-t001:** Features of the nine studies included in this systematic review and meta-analysis.

Authors	Year	Site	Fetuses with Recurrent Anomalies (*N*)	Specific Recurrent Anomalies Included	Clinical ES (CES) or Whole ES (WES)	Sanger Validation
Alamillo et al. [19]	2015	Aliso Viejo, CA, USA	6	No	WES-trio	Yes
Yates et al. [20]	2016	Gaithersburg, MD, USA	24	No	CES	Yes
Pangalos et al. [21]	2016	Athens, Greece	3	No	CES	Yes
Meier et al. [22]	2018	Basel, Switzerland	8	No	WES-trio	Yes
Borrell et al. [28]	2019	Barcelona, Catalonia, Spain	12	Yes	CES-solo	Yes
Corsten-Janssen et al. [23]	2019	Groningen, Netherlands	4	No	CES-trio	No
Guo et al. [16]	2019	Beijing, China	40	Yes	WES-trio	Yes
Greenbaum et al. [24]	2019	Tel Hashomer, Israel	15	No	WES-trio, quatro & solo	No
Vora et al. [25]	2020	Chapel Hill, NC, USA	28	No	CES-trio	Yes

CES: clinical exome sequencing; WES: whole-exome sequencing.

**Table 2 jcm-10-04739-t002:** Eighty-seven causative variants found in the 57 fetuses with recurrent structural anomalies and a positive ES result included in this systematic review.

Author	Phenotype	Gene	Variant	Type of Variant	Classification of Variant	Inheritance	Zygosity	Syndrome or Disease
**Multisystem Multiple Anomalies**								
Alamillo et al. [19]	Omphalocele, cleft lip and palate	*OFD1*	c.929T>C	Missense	L.Pat.	XLR	Hemizygous	Oral-facial-digital syndrome 1
Alamillo et al. [19]	Edema, small and bell-shaped chest, and scalloping of the ribs	*RAPSN*	c.484G>A	Missense	Pat.	AR	Heterozygous	RAPSN-associated fetal akinesia deformation sequence
Yates et al. [20]	Hydrops, contractures, and echogenic kidney	*FOXP3*	c.1009C>T (p.R337X)	Nonsense	Pat	XLR	Hemizygous	IPEX syndrome
Yates et al. [20]	Macrocephaly, hydrocephalus, cleft lip and palate, cardiac defect, and bifid thumb	*AMER1*	c.705delT	Frameshift	Pat	XLR	Hemizygous	Osteopathia striata with cranial sclerosis
Meier et al. [22]	1st: Cerebral hypoplasia, cerebellar hypoplasia, agenesis of occipital lobes, bilateral renal agenesis, ureteral agenesis, and uterine hypoplasia. 2nd: Corpus callosum agenesis, cerebral hypoplasia, arhinencephaly, bilateral renal hypoplasia and cystic dysplasia, ureteral hypoplasia, uterine hypoplasia, and vaginal atresia.	*KIF14*	NM014875.2: c.1750_1751delGA;1780A>T	Frameshift; missense	Pat.; Pat.	AR	Compound Heterozygous	Isolated microcephaly
Guo et al. [16]	Hydrocephalus, hydrops	*L1CAM*	NM_000425.5: c:3581C>T	Frameshift	L.Pat.	XLR	Hemizygous	Hydrocephalus due to aqueductal stenosis
Guo et al. [16]	Hydrops, intestinal obstruction, and polyhydramnios	*SLC26A3*	NM_000111.3: c.2006C>A	Nonsense	Pat.	AR	Homozygous	Congenital chloride diarrhea
Guo et al. [16]	Encephalocele and polycystic kidney dysplasia	*CC2D2A*	NM_001080522: c.1751G>A; c.3293T>G	nonsense	Pat.; Pat.	AR	Compound Heterozygous	Meckel syndrome 6
Guo et al. [16]	Anencephaly, heart defect, and polyhydramnios	*PUS3*	NM_031307.4: c.838C>T; c.340T>C	Nonsense; missense	L.Pat.; L.Pat.	AR	Compound Heterozygous	Mental retardation autosomal recessive 55
Guo et al. [16]	Hydrops, intestinal obstruction, and polyhydramnios	*SLC26A3*	NM_000111.3: 269_270dup; c.1000G>T	Frameshift; nonsense	Pat.; Pat.	AR	Compound Heterozygous	Congenital chloride diarrhea
Guo et al. [16]	Encephalocele, hydrocephalus, and polycystic kidney dysplasia	*CEP290*	NM_025114.3: c.613C>T; c.5329C>T	Nonsense; nonsense	Pat.; Pat.	AR	Compound Heterozygous	Meckel syndrome 4
Guo et al. [16]	Hydrocephalus, arthrogryposis multiplex, and talipes	*KIAA1109*	NM_015312.3: c.692del; c.3323+1G>A	Frameshift; splicing	L.Pat; L.Pat.	AR	Compound Heterozygous	Alkuraya–Kucinskas syndrome
Guo et al. [16]	Renal agenesis, hemivertebrae, and right aortic arch	*KIAA1109*	NM_015312.3: c.9153del; c.13849+11G>C	Frameshift; splicing	Pat.; VUS	AR	Compound Heterozygous	Alkuraya–Kucinskas syndrome
Guo et al. [16]	Deformed rib cage, short ribs, short long bones, cardiac defect, and abnormal lung	*DYNC2H1*	NM_001080463.2: c.9929T>C; c.5920G>T	Missense; nonsense	VUS; L.Pat.	AR	Compound Heterozygous	Short-rib thoracic dysplasia 3 with or without polydactyly
Guo et al. [16]	Aplasia/ hypoplasia of the fibula, ankle contracture	*C2CD3*	NM_015531.6: c.3741G>C; c.159_160insC	Missense; frameshift	L.Pat.; L.Pat.	AR	Compound Heterozygous	Orofaciodigital syndrome XIV
Guo et al. [16]	Cardiac defect, short long bones, and cystic hygroma	*CO11A2*	NM_080679.2 c.1773+8T>A; c.971dup	Splicing; frameshift	VUS; L.Pat.	AR	Compound Heterozygous	Fibrochondrogenesis
Guo et al. [16]	Holoprosencephaly, hydrocephalus, median cleft lip and palate, and cardiac defect	*ZRSR2*	NM_005089.3: c.1207_1208del	Frameshift	L.Pat.	XLD	Hemizygous	X-linked intellectual disability
Guo et al. [16]	Encephalocele, dysgenesis of the cerebellar vermis, polydactyly, and median cleft lip and palate	*C5ORF42*	NM_023073.3: c.3707delinsTT; c.7993_7994del	Frameshift; frameshift	L.Pat.; L.Pat.	AR	Compound Heterozygous	Orofaciodigital syndrome VI
Guo et al. [16]	Ventriculomegaly, ambiguous genitalia	*MAGEL2*	NM_019066.5: c.1996del	Frameshift	Pat.	AD	Heterozygous	Schaaf–Yang syndrome
Greenbaum et al. [25]	1st. Fetal akinesia, mild polyhydramnios, small stomach, suspected right club foot, extended lower limbs, clenched hands, and neck hyperextension. 2nd: Arthrogryposis, hypotonic features, and abnormal posture.	*LMOD3*	NM_198271: c.723_733del; c.360dupA	Frameshift; frameshift	-	AR	Compound Heterozygous	Nemaline Myopathy 10
Greenbaum et al. [24]	1st: Abnormal spine and chest, unusual skull shape, and echogenic cystic and horseshoe-like kidneys. 2nd: Increased NT, generalized edema, spine distortion, bilateral clubfoot, and absent nasal bones. 3rd: Reduced/lack ossification in the skull, ribs and vertebrae, protruding abdomen, and short trunk.	*BMPER*	NM_133468.5: c.410T>A	Missense	-	AR	Homozygous	Diaphanospondylodysostosis
Greenbaum et al. [24]	1st: Encephalocele, large multicystic kidneys, oligohydramnios, and lack of urinary bladder and stomach demonstration. 2nd: Posterior fossa abnormality, short and malformed corpus callosum, and IUGR; SUA, small dysgenic kidney, urinary bladder was not visualized, oligohydramnios, and hypertelorism.	*TCTN2*	NM_024809.4: c.1506-2A>G	Splicing	-	AR	Homozygous	Meckel syndrome 8
Greenbaum et al. [24]	1st: Posterior urethral valve, cystic kidney finding, and suspected omphalocele. 2nd: Increased NT, cystic lesion near umbilical cord insertion site. 3rd: Cystic, hygroma, partial vermian agenesis, ARSA, omphalocele, and echogenic and multicystic kidneys. 4th: Increased NT, facial dysmorphism, echogenic kidneys, omphalocele, post-axial polydactyly clubfoot, and cardiac defect.	*PIGN*	NM_176787.5: c.163C>T; NM_176787.5: c.2283G>C	Nonsense; missense	-	AR	Compound Heterozygous	Multiple congenital anomalies-hypotonia-seizures syndrome 1
Greenbaum et al. [24]	1st: Large polycystic kidney, oligohydramnios, and moderate bilateral ventriculomegaly. 2nd: Polycystic kidneys, hydrocephalus, megacisterna magna, and macrocephaly. 3rd: Enlarged echogenic kidneys, severe oligohydramnios hydrocephalus, megacisterna magna, and thin corpus callosum	*CPT2*	NM_001330589.1: c.1239_1240del	Frameshift	-	AR	Homozygous	CPT II deficiency, lethal neonatal
Vora et al. [25]	Ventriculomegaly, cystic kidneys, anhydramnios, cardiac defect, and bilateral polydactyly (*n* = 4)	*CEP290*	c.384_387 delTAGA; (p.Asp128Glufs); c.1936 C>T (p.Gln646Ter)	Frameshift; missense	Pat.; Pat.	AR	Compound Heterozygous	Meckel syndrome 4
Vora et al. [25]	Hand and foot clefting, syndactyly, facial clefting, and renal pyelectasis (*n* = 3)	*TP63*	c.1028G>C (p.Arg343Pro)	Missense	Pat.	AD	Heterozygous	Ectrodactyly, ectodermal dysplasia, and cleft/lip
Vora et al. [25]	Renal agenesis and heart defect	*GREPB1L*	c.4881_4882delCA (p.H1627f); c.277G>A (p.E92K)	Frameshift: missense	L.Pat.; VUS	AR	Compound Heterozygous	Renal hypoplasia/aplasia
Vora et al. [25]	Cystic hygroma, hydrops, complex heart defect	*FOXC2*	c.612delC (p.Pro204fs)		L.Pat.	AD	Heterozygous	Lymphedema–distichiasis syndrome
Vora et al. [25]	Sloping forehead, micrognathia, brachycephaly, bilateral, ribs appear flared, short long bones, ambiguous genitalia, and contractures of hands and feet bilaterally	*TRAIP*	c.140 C>T(p.P47L); c.553 C>T(p.R185Ter)	Missense	VUS; Pat.	AR	Compound Heterozygous	Seckel syndrome 9
Vora et al. [25]	Agenesis corpus callosum, shortened long bones, arthrogryposis, suspected tetralogy of Fallot, micrognathia, hypertelorism, kyphoscoliosis, ambiguous genitalia, and rocker bottom feet (*n* = 2)	*ALG3*	c.518C>T (p.R163C); c.1185G>C (p.R385T)	Missense	VUS; VUS	AR	Compound Heterozygous	Congenital disorder of glycosylation, 1D
Vora et al. [25]	Enlarged bladder with distorted abdomen, extremely short long bones and small chest, bilateral polydactyly on hands, and neck fixed in a flexed position. Previous pregnancies have also shown thick NT/cystic hygroma, and enlarged cisterna magna with possible ventriculomegaly.	*TRAF3iP1*	c.169G>A (p.Glu57Lys); c.988-1G>C	Splicing	VUS: L.Pat.	AR	Compound Heterozygous	Gene typically Senior Loken syndrome 9 but this is a more severe presentation. Ciliopathy.
**Central Nervous System Anomalies**								
Yates et al. [20]	Hydrocephalus (aqueductal stenosis)	*L1CAM*	c.2087delG	Frameshift	Pat	XLR	Hemizygous	Hydrocephalus
Pangalos et al. [21]	Hydrocephalus +FGR	*NEB*	NM_004543.5: c.11060C>T; c.11333T>C	Canonical missense; missense	L.Pat.; L.Pat.	AR	Compound Heterozygous	Nemaline myopathy (OMIM 2560) (AR)
Pangalos et al. [21]	Brain MRI abnormalities	*ASS1*	NM_000050: c.725C>T; c.971G>T	Missense	L.Pat.; Pat.	AR	Compound Heterozygous	Citrullinemia (OMIM 215700) (AR)
Meier et al. [22]	Meckel–Gruber syndrome like	*MSK1*	NM_017777.3:c.417G>A	Splicing	L.Pat.	AR	Homozygous	Meckel–Gruber syndrome
Meier et al. [22]	Dandy–Walker malformation	*PIGW*	NM178517: c.106A>G	Missense	L.Pat.	AR	Homozygous	Glycosylphosphatidylinositol biosynthesis defect 11
Corsten-Janssen et al. [23]	Cerebellar vermis hypoplasia, hydronephrosis	*PEX1*	NM_000466.2:c.2097dupT	Frameshift	-	AR	Homozygous	Zellweger syndrome
Guo et al. [16]	Dysgenesis of the cerebellar vermis	*POMT1*	NM_007171.3: c.110_113dup; c.169C>T	Frameshift; nonsense	Pat.; Pat.	AR	Compound Heterozygous	Muscular dystrophy dystroglycanopathy type A
Greenbaum et al. [24]	Occipital encephalocele, ventriculomegaly	*B3GALNT2*	NM_001277155.2: c.236-1G>C	Splicing	-	AR	Homozygous	Muscular dystrophy–dystroglycanopathy
Corsten-Janssen et al. [23]	Severe hydrocephaly	*POMGNT1*	NM_001243766.1: c.636C>T	Synonymous	-	AR	Homozygous	Walker–Warburg syndrome
Borrell et al. [28]	Lissencephaly	*ASPM*	NM_018136.4: c.7551T>G c.9279G>A	Nonsense; nonsense	L.Pat; Pat.	AR	Compound Heterozygous	Microcephaly with simplified gyral pattern
**Fetal Hydrops**								
Alamillo et al. [19]	Hydrops	*GBE1*	c.1064G>A; c.1543C>T	Missense; missense	L.Pat.; Pat	AR	Compound Heterozygous	Glycogen storage disease IV
Guo et al. [16]	Hydrops	*RAPSN*	NM_032645.5: c.969C>A; c.149_153delinsGATGGGCCGCTACAAGGAGATGG	Nonsense; frameshift	Pat.; Pat.	AR	Compound Heterozygous	Fetal akinesia deformation sequence 2
Guo et al. [16]	Hydrops	*RYR1*	NM_001042723.2: c.2286del; c.6721C>T	Frameshift; nonsense	Pat.; Pat.	AR	Compound Heterozygous	Multiple pterygium syndrome lethal type
Guo et al. [16]	Hydrops fetalis	*PIEZO1*	NM_001142864.4: c.1536_1537del; c.4610_4617dup	Frameshift; frameshift	L.Pat; L.Pat.	AD/AR	Compound Heterozygous	Lymphatic malformation 6
Vora et al. [25]	Hygroma	*FOXC2*	c.251C>T (p.Ala84Val)		L.Pat.	AD	Heterozygous	Lymphedema-distichiasis syndrome
Borrell et al. [28]	Hydrops	*SEC23B*	NM_001172745: c.716A>G	Missense	L.Pat.	AR	Homozygous	Congenital dyserythropoietic anemia type II
Guo et al. [16]	Hydrops	*FOXP3*	NM_014009.4: c.1120_1122del	In frame	L.Pat.	XLR	Hemizygous	Immunodysregulation, polyendocrinopathy and enteropathy, and X-Linked.
**Musculoskeletal Anomalies**								
Guo et al. [16]	Hemivertebrae	*DLL3*	NM_016941.4: c.661C>T	Nonsense	Pat.	AR	Homozygous	Spondylocostal dysostosis 1
Guo et al. [16]	Multiple joint contractures	*GLDN*	NM_181789.4: c.1240C>T; c.1027G>A	Nonsense; missense	Pat.; L.Pat.	AR	Compound Heterozygous	Lethal Congenital Contracture Syndrome 11
Guo et al. [16]	Ankle contracture, arthrogryposis multiplex, scoliosis.	*CHRNG*	NM_005199.5: c.13C>T; c.202C>T	Nonsense; nonsense	L.Pat.; L.Pat.	AR	Compound Heterozygous	Multiple pterygium syndrome lethal type
Greenbaum et al. [24]	1st: Distal arthrogryposis (hands). 2nd: Bilateral clubfoot.	*FKBP14*	NM_017946.3: c.568_570del	In frame	-	AR	Homozygous	Ehlers–Danlos syndrome, kyphoscoliotic type, 2
Greenbaum et al. [24]	1st: Short long bones, IUFD. 2nd: Narrow thorax, bowed femur, short long bones.	*EVC2*	NM_147127.4: c.572A; c.3265C>T	Missense; nonsense	-	AR	Compound Heterozygous	Ellis–Van Creveld syndrome
Vora et al. [25]	Arthrogryposis (*n*=3). One with dextrocardia, partial agenesis of the right lung.	*ADGRG6/GPR126*	c.2515C>T (p.His839Tyr)	Missense	VUS	AR	Homozygous	Lethal Congenital Contracture Syndrome 9
Borrell et al. [28]	Arthrogryposis multiplex	*DOK7*	NM_001164673: c.230C>T; NM_173660: c.532+4A>G	Missense; non coding variant	L.Pat.; L.Pat.	AR	Compound Heterozygous	Fetal akinesia deformation sequence
**Cardiac Defects**								
Guo et al. [16]	Cardiac defect	*NODAL*	NM_018055: c.823C>T; c.172_174del	Missense; in frame	Pat.; VUS	AD/AR	Compound Heterozygous	Heterothaxis visceral 5
Guo et al. [16]	Cardiac defect	*NONO*	NM_001145408.2: c.246_249del	Frameshift	L.Pat.	XLD	Hemizygous	Mental retardation autosomal recessive

AR: Autosomal Recessive; AD: Autosomal Dominant; XLD: X-linked Dominant, XLR: X-linked Recessive.

## Data Availability

All data generated are included in this article and its Appendix A. Further enquiries can be directed to the corresponding author.

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
