# Peer review of "Prenatal Exome Sequencing in Recurrent Fetal Structural Anomalies: Systematic Review and Meta-Analysis"

_jcm, 2021, doi:10.3390/jcm10204739_

Round 1

Reviewer 1 Report

The Authors performed a systematic review and meta-analysis on the use of exome sequencing techniques in recurrent fetal structural abnormalities. The topic is very interesting and the work is very well designed and well conducted. The results provide an advance in current knowledge, as the contribution is useful for diagnostic choices to be made in pregnancy by all readers involved in prenatal ultrasound and genetic diagnosis. The manuscript is well written and the presentation of the data needs only minor improvements.

Some minor remarks:

On page 1, Introduction, perhaps the word "nowadays" is repeated too many times.

The study's overall results should be summarized in a table, which would make it easier for the reader to benefit from reading the article, given the plenty of numbers and percentages in the text.

In table 2, gene names are correctly shown in italics. The same should be done in the text of the article.

The first paragraph on page 2 (lines 47-53) is a little difficult to read and there is an apparent inconsistency between the risk of recurrence given in the first part of the paragraph (4.08%) and the later statement that the absolute risk is 2%.

On pages 5 and 6, Monogenic Variants, there is some confusion between variants and cases. Initially, it says that 45 variants (79%) were found in genes with autosomal recessive inheritance, then it says that "Among the 45 cases described with autosomal recessive inheritance, homozygosity was identified in 13 (29%) cases".

The same confusion is on page 10, Discussion, lines 211-213.

In table 2, the separation between the cases of the various authors is not clear (review the use of the longer horizontal lines). The citation should also include the reference number.

On page 10, lines 223-224, the sentence "some authors consider polydactyly as the third criterion as a frequent feature" should be revised.

On page 12, reference 15 should be corrected.

Finally, a personal observation: in the manuscript, the microarray examination technique is referred to as “chromosomal microarray (CMA)”. This definition is widely used in the literature, but in my personal opinion it is absolutely wrong. This test should be better defined as "genomic microarray (GMA)" as it gives information on the possible presence of genomic imbalances, without telling us anything about the chromosome structure. To be clearer: we can have a normal result with the microarray and the subject may have an altered chromosome constitution (e.g. balanced rearrangements or even unbalanced rearrangements, but involving regions not covered by the array). The reverse can also occur, i.e. the microarray shows a genomic imbalance and the standard karyotype is normal. In any case, the microarray, when it shows an abnormality, tells us whether a certain genomic tract is duplicated or deleted, but it does not give us any information about the chromosomes involved, it does not tell us, for example, whether a duplication is a simple duplication or the result of an insertional translocation, which is extremely important because of the risk of recurrence.

Author Response

Thank you for the opportunity to improve our paper. We have addressed all the reviewers’ comments point by point

-On page 1, Introduction, perhaps the word "nowadays" is repeated too many times.
Changes made accordingly

-The study's overall results should be summarized in a table, which would make it easier for the reader to benefit from reading the article, given the plenty of numbers and percentages in the text.

We assume that the Figure 3 is summarizing the main data of the study and in one sentence of the Abstract A pathogenic or likely pathogenic variant was found in 57 fetuses, resulting in a 40% (95%CI: 26% to 54%)”

-In table 2, gene names are correctly shown in italics. The same should be done in the text of the article.

Changes made accordingly

-The first paragraph on page 2 (lines 47-53) is a little difficult to read and there is an apparent inconsistency between the risk of recurrence given in the first part of the paragraph (4.08%) and the later statement that the absolute risk is 2%.

The paragraph has been rewritten for a better understanding: “It has been shown that, for women whose first pregnancy was affected by a fetal structural anomaly, the absolute risk of recurrent congenital anomaly in the second pregnancy was 4.08%, 2% for a similar anomaly plus 2% for a dissimilar anomaly. Regarding the recurrence risk (RR), it was considerably elevated (RR = 24) for a similar anomaly, while for dissimilar anomalies the increase was more modest (RR = 1.4)”

-On pages 5 and 6, Monogenic Variants, there is some confusion between variants and cases. Initially, it says that 45 variants (79%) were found in genes with autosomal recessive inheritance, then it says that "Among the 45 cases described with autosomal recessive inheritance, homozygosity was identified in 13 (29%) cases".

We have corrected this mistake: “In 45 cases (79%) causative variants were found…”

The same confusion is on page 10, Discussion, lines 211-213.

We have clarified that sentence: “In most of the positive cases (79%) an autosomal recessive inheritance was identified”

-In table 2, the separation between the cases of the various authors is not clear (review the use of the longer horizontal lines). The citation should also include the reference number.

Changes made in accordance

-On page 10, lines 223-224, the sentence "some authors consider polydactyly as the third criterion as a frequent feature "should be revised.

We have changed the definition of Meckel syndrome to: “Meckel syndromre is characterized by the triad of cystic renal disease, posterior fossa abnormalities (usually occipital encephalocele), and the hepatic ductal plate malformation leading to hepatic fibrosis and bile duct proliferation. Polydactyly is relatively common.”

-On page 12, reference 15 should be corrected.

We have corrected this mistake.

-Finally, a personal observation: in the manuscript, the microarray examination technique is referred to as “chromosomal microarray (CMA)”. This definition is widely used in the literature, but in my personal opinion it is absolutely wrong. This tests hould be better defined as "genomic microarray (GMA)" as it gives information on the possible presence of genomic imbalances, without telling us anything about the chromosome structure. To be clearer: we can have a normal result with the microarray and the subject may have an altered chromosome constitution (e.g. balanced rearrangements or even unbalanced rearrangements, but involving regions not covered by the array). The reverse can also occur, i.e. the microarray shows a genomic imbalance and the standard karyotype is normal. In any case, the microarray, when it shows an abnormality, tells us whether a certain genomic tract is duplicated or deleted, but it does not give us any information about the chromosomes involved, it does not tell us, for example, whether a duplication is a simple duplication or the result of an insertional translocation, which is extremely important because of the risk of recurrence.

We agree with the considerations of the reviewer and we have change CMA by “microarray” since this is the term used in ISCN 2020

Reviewer 2 Report

Review and meta-analysis in this manuscript showed diagnostic yield of WES/CES in diagnostic of reccurent/non recuurent fetal anomalies. 

As it can be ssen from the manuscript the various type of anomalies was present in foetusus as well as various other conditions (hydrops). If it is possible it would be desirable determine diagnostic yield for speciffic anomalies (brain anomalies, kidney anoamlies) and conditions ( hydrops). 

The diagnostic yield raged from 12-67 % and it should be explained in discussion more clearly. 

Author Response

Thank you for the opportunity to improve our paper. We have addressed all the reviewers’ comments point by point

-As it can be ssen from the manuscript the various type of anomalies was present in foetusus as well as various other conditions (hydrops). If it is possible it would be desirable determine diagnostic yield for speciffic anomalies (brain anomalies, kidney anoamlies) and conditions ( hydrops).

The diagnostic yield raged from 12-67 % and it should be explained in discussion more clearly.

We have added a sentence in the Results “When the 57 positive cases were grouped according the anatomic systems that were involved in the recurrent structural anomalies, the most commonly observed pattern was multisystem anomalies (n=30), followed by central nervous system (n=11), hydrops (n=7), musculoskeletal (n=7), and cardiac anomalies (n=2).” and in the Discussion “Another reason that can explain the wider range of diagnostic yields (12% to 67%) observed among the included series is the different type of recurrent structural anomalies included. Multisystem anomalies appear to carry a higher diagnostic yield since they account for more than half of the positive cases.”